# Epigenetic Effects of Polybrominated Diphenyl Ethers on Human Health

**DOI:** 10.3390/ijerph16152703

**Published:** 2019-07-29

**Authors:** Robert G. Poston, Ramendra N. Saha

**Affiliations:** Molecular and Cell Biology Department, School of Natural Sciences, University of California, Merced, 5200 North Lake Road, Merced, CA 95343, USA

**Keywords:** polybrominated diphenyl ethers (PBDE), neurodevelopment, epigenetics, DNA methylation, chromatin remodeling, environmental toxins, toxicity

## Abstract

Disruption of epigenetic regulation by environmental toxins is an emerging area of focus for understanding the latter’s impact on human health. Polybrominated diphenyl ethers (PBDEs), one such group of toxins, are an environmentally pervasive class of brominated flame retardants that have been extensively used as coatings on a wide range of consumer products. Their environmental stability, propensity for bioaccumulation, and known links to adverse health effects have evoked extensive research to characterize underlying biological mechanisms of toxicity. Of particular concern is the growing body of evidence correlating human exposure levels to behavioral deficits related to neurodevelopmental disorders. The developing nervous system is particularly sensitive to influence by environmental signals, including dysregulation by toxins. Several major modes of actions have been identified, but a clear understanding of how observed effects relate to negative impacts on human health has not been established. Here, we review the current body of evidence for PBDE-induced epigenetic disruptions, including DNA methylation, chromatin dynamics, and non-coding RNA expression while discussing the potential relationship between PBDEs and neurodevelopmental disorders.

## 1. Introduction 

Across the world, humans face exposure to a vast number of industrial chemicals, whose potential for negatively impacting human health has long been a concern [1,2,3]. In early 2018, the United States Environmental Protection Agency (EPA) reported 30,972 active chemicals in industry out of a total of 86,071 registered in the agency’s Toxic Substances Control Act (TSCA) Chemical Substance Inventory. The European Chemicals Agency’s (ECHA) most recently updated figure from their relatively new Regulation for Registration, Evaluation, Authorization and Restriction of Chemicals (REACH) initiative reports 21,403 unique substances. China has also established a program recently—similar to Europe’s REACH regulations—that mandates new updating of China’s chemical inventory, the Inventory of Existing Chemical Substances (IECSC), which lists 45,612 substances as of 2013. As world governments attempt to define what chemicals have been produced and are in use, efficient methods to identify and evaluate compounds for safety screening are still being debated and formed. Progress is slow, with few chemicals actually being heavily regulated. In the US, the history of chemical regulation is long and convoluted, and is well reviewed elsewhere [4]. Presently, the EPA is in the midst of a three-tiered evaluation program designed to assess the safety of existing chemicals, with only the most dangerous chemicals likely to ever reach the eventual ‘Risk Management’ phase. It is questionable whether this type of approach is practical at all, yet meaningful change may not come soon, as it is unlikely that the country will shift the burden of proof regarding chemical safety from regulatory agencies to manufacturers (as with Europe’s REACH program). In the meantime, the vast volume and diversity of industrial chemicals we expose ourselves to continues to pose a potentially serious risk to human health. There are numerous avenues by which hazardous compounds may impact human health, perhaps the most widely recognized of which are potential for carcinogenicity, adverse effects on reproductive health, and disruption of hormonal signaling. Another exceedingly concerning endpoint for human health is nervous system toxicity, particularly during development of the brain. The developing brain is an especially vulnerable target due to the complex nature of its formation and refinement that spans prenatal and years of postnatal development. As such, neurodevelopmental toxicity induced by chemical exposures has been heavily studied [5], but much remains unclear. Here, we will focus on a class of industrial chemicals that has been under heavy scrutiny for suspected neurodevelopmental toxicity: polybrominated diphenyl ethers (PBDEs).

PBDEs are a group of environmentally persistent chemicals that have been widely used as flame retardants on household consumer products since 1970s. Due to their environmental stability and propensity for bioaccumulation, PBDE concentrations have ubiquitously and cumulatively built up in our environment and in our bodies around the globe. Intriguingly, PBDEs enter the environment from both anthropogenic and natural sources. Historically, these compounds were first described in the biomedical literature as early as the 1960s—a decade before their anthropogenic production—when they were isolated from Australian marine sponges (*Dysidea sp.*) and found to have antimicrobial properties [6,7,8]. They have also been isolated from various red algae [9,10]. Recently, in the case of sponges, it was demonstrated that PBDEs are actually produced by symbiotic cyanobacteria and are theorized to confer some level of microbial resistance to the host sponges, although the mechanism(s) by which the compounds are toxic to other organisms remains unknown [11]. It is interesting to note, however, that these compounds are excreted by the cyanobacteria and subsequently accumulate in high concentrations, crystalizing in the sponge ectosomal tissues. This is perhaps how sponges avoid the compounds’ toxic effects and how they may be a defense mechanism against potential eukaryotic predators such as fish in addition to other prokaryotes [12]. In the context of human health, it is unfortunate that such a class of compounds, whose natural production was likely evolutionarily driven by their toxicity, ended up becoming a flame retardant of choice for consumer products. Understanding the natural origins of PBDEs may also inform our investigation of their biological effects in humans, which is of pressing importance given another unfortunate aspect of PBDE biology—the growing evidence for their epidemiological association with neurodevelopmental disorders (NDDs). In this review, we will briefly discuss known biological mechanisms affected by PBDEs, focusing on epigenetic impairments and the impacts these disruptions may have on human health, especially in the context of neurodevelopmental disorders.

## 2. Methods

Records were initially identified by searching PubMed and Web of Science databases for combinations of the terms: ‘PBDE’, ‘BDE’, ‘polybrominated diphenyl ethers’, ‘chromatin’, ‘methylation’, ‘DNA methylation’, ‘histone’, ‘histone modification’, and ‘non-coding RNA’. In total, 122 records were identified from the searches. We found 38 unique records after removal of duplicates and full text screening for studies that specifically reported epigenetic endpoints (chromatin integrity, DNA methylation, histone expression/histone modifications, non-coding RNA expression). All included studies were used for qualitative analysis. No quantitative meta-analysis was conducted. Figure 1 was created with BioRender. 

## 3. Relation of PBDEs to Human Health

### 3.1. Human Exposure to PBDEs and Effects on Human Health, Especially Neurodevelopmental Disorders

A major effort has been devoted to evaluating the potential risk of PBDE exposure to human health. For the sake of brevity, here, we will highlight some of the major points while pointing to relevant reviews and meta-analyses of the vast number of studies that have been published on the topic. Monitoring of environmental and human levels of PBDEs in the 1990s led to a rising general concern that they may be a serious human health risk, which was widely recognized by the early 2000s [13]. At the time, the extents of compounds’ toxicity were unclear despite the observed increasing toxicological evidence. Following much attention in the time since, it is now clear that exposure to PBDEs is a very real concern for humans around the world as the compounds are environmentally stable and lipophilic, and thus tend to bioaccumulate and also collect in households, primarily in dust. These routes of accumulation enable the most common modes of human exposure—primarily through ingestion and inhalation of dust [14,15] and dietary intake, predominantly from seafood and dairy products [16]. It is also concerning and of interest that infants and toddlers tend to have higher body burdens compared to adults when considering potential developmental toxicity [17]. This is thought to be caused by younger children having higher rates of intake from dust and household products, as well as by additional exposure to PBDEs through breastmilk. 

Due to the persistent, widespread, and sometimes heavy exposure levels observed, much attention has been given to the roles of PBDEs in several major aspects of health: carcinogenicity, reproductive health, and disruption of hormonal signaling [16,18,19]. In addition to these concerning PBDE-related effects on human health, another serious worry is their neurotoxicity and potential roles in the etiology of neurodevelopmental disorders. A substantial amount of work has been done, surveying the potential association of PBDE exposures with behavioral deficits in humans, as well as in other animal models. Recently, several large-scale and systematic reviews have been conducted, both of evidence from human [20,21,22] and animal studies [23]. Briefly, they conclude that PBDE exposures highly correlate with externalizing behaviors and IQ in children, while BDE-47/99/209 were concluded to affect learning in animal studies. There is also of concern for the relationship between environmental toxins such as PBDEs and autism spectrum disorders, although the relationship is less clear, especially in human studies [24]. Given the established and suspected connections between PBDE exposures and intelligence and behavioral deficits, as well as other aspects of human health, it is imperative to strive for a mechanistic understanding of PBDE toxicity at the molecular and cellular level (Figure 1).

### 3.2. Biological Mechanisms of PBDE Toxicity

Since the rise of concern regarding PBDE toxicity, several major impacted biological mechanisms have been identified and investigated. These and other less explored effects of PBDEs have been recently reviewed [25]. Briefly, major identified points of toxicity are: (1) disruption of calcium signaling—dating back to one of the earliest functional studies of PBDEs [26]; (2) interference with thyroid hormone homeostasis—thought to be enabled by structural similarity to the hormones; (3) general cellular toxicity driven by mitochondrial disruption and elevated production of reactive oxygen species (ROS), in some cases leading to DNA damage and apoptosis. However, some of the specifics of these known effects remain unresolved and other unknown mechanisms may be involved. Additionally, most studies have focused on individual PBDEs, while there are 209 congeners that humans are exposed to in mixtures, and these compounds can be further processed to produce various metabolites. Such metabolism, thought to be endogenously mediated in humans by cytochrome P450 enzymes [27,28,29] and inherent in some other natural contexts [9,10,11,30], leads to the production of hydroxylated and methoxylated forms that await in-depth investigation. Additional confounding factors include that past studies have covered a wide range of concentrations, many likely higher than environmentally relevant exposures, and much of this work has been conducted in cell culture, requiring further in vivo confirmation. Thus, while an extensive amount of work has been done, there is much that may yet be uncovered regarding biological mechanisms of PBDE toxicity.

A recently expanding approach towards understanding PBDE toxicity has focused on PBDE-induced disruption of epigenetic regulation. Such mechanisms are interesting as potential PBDE targets as they would constitute a direct gene-environment platform for cellular disruption. Further, there is an expanding appreciation for the role of epigenetic mechanisms in neurodevelopment and cognition [31,32], as well as in diseases of the nervous system, including neurodevelopmental disorders [33,34]. Therefore, it may prove useful to understand the effects of PBDE exposures on epigenetic components, both in the brain and across other cell and tissue types, in order to build more complete causal models towards explaining observed links to human health complications and behavioral deficits. However, compared to other widely studied mechanisms, relatively little attention has been given to epigenetic effects of PBDEs. Here, we will summarize the findings of studies conducted to date that have observed epigenetic endpoints such as DNA methylation, chromatin characteristics including modifications and remodeling of histones, and other epigenetic mechanisms such as expression of various non-coding RNAs.

### 3.3. Repeatedly Observed Disruption of DNA Methylation

One of the most commonly recognized epigenetic mechanisms is DNA methylation. Methylation commonly occurs at cytosine nucleotides positioned before a guanosine (CpG dinucleotides), resulting in a 5-methylcytosine. Given the extensive impact methylation has on transcriptional regulation [35], it comes as little surprise that its disruption has potential impacts on human health [36]. An interesting example is folate deficiency’s implication in disrupted methylation during pre-natal development, although the relationship is incompletely understood [37]. The relationship between PBDE exposure and DNA methylation is similarly incompletely understood. Although most studies report some correlation, they do not have a clear consensus, especially in human samples, while in vitro studies more consistently report negative correlations (see references below). 

Studies have assessed PBDE-exposure-dependent changes in global DNA methylation at various representative regions or at specific loci (promoters). Two of the most prominent examples of representative methylated regions include ALU elements and LINE1, which are common transposable repeats that can have adverse cellular impacts when de-repressed due to hypomethylation [38]. Repetitive elements make up a large portion of the human genome [39] and have high CpG frequency, contributing heavily to the global amount of DNA methylation and thereby serving as a reasonable global estimate. 

One of the earliest studies on the effects of PBDE exposure on global DNA methylation in humans found a negative relationship between measured BDE-47 levels and ALU %5mC in blood samples of healthy Korean adults, while not finding significant relationships for BDE-99 or LINE1 methylation [40]. Similar studies correlating PBDE levels in blood with methylation have followed. One found an inverse relationship between BDE-47 abundance and TNFα promoter methylation in cord blood samples from mother–infant pairs of the Boston Birth Cohort [41]. Another reports a more complex finding in newborn cord blood samples from the CHAMACOS study, wherein significant changes in LINE1 methylation were found when considering co-exposure to DDT, DDE, and PBDEs (the direction of change depended on level of DDE or DDT co-exposure) [42].

Several groups have also examined the relationship between PBDE levels and effects on the placental epigenome. In 2016, two reports were published on effects in human placental samples. In one, the authors made PBDE, PCB, DDE measurements in villous placental tissue samples and found positive associations of PBDE levels with IGF2/H19 imprinting and methylation status (bisulfite conversion and targeted pyrosequencing) and global DNA methylation (assessed by LUMA (luminometric methylation assay)) [43]. In the other, PBDE levels in umbilical cord blood were measured from eighty human samples and correlated with placental DNA methylation levels in LINE1, NR3C1, and IGF2. BDE-66/153/209 were all found to have significant negative correlations with methylation of some of these loci [44]. Two very recent reports have also been made utilizing in vitro models of the placenta. One group exposed primary villous cytotrophoblasts (CTBs, an in vitro model of human placental development) to BDE-47 or BDE-99. They found that BDE-47 alters gene expression in a concentration-dependent manner and produced a low-level global increase in DNA methylation (assessed with HumanMethylation450 beadarray) [45]. Another group exposed human placental choriocarcinoma cells (BeWo cells) to 1 uM BDE-47 and found reduced methylation of some CpG loci of mitochondrial biomarkers (with no differences found for 50 uM exposures) [46].

In addition to these human studies, PBDE-methylation relationships have also been investigated in model animals—mostly rodents—both in vitro and in vivo. In vitro studies have been conducted in different cell types, but consistently found negative correlations between PBDE exposure and methylation level. In the earliest of these studies, primary hippocampal neurons were exposed to various concentrations of BDE-209 for 24 hours and subsequently, a global decrease in DNA methylation was found by an antibody based ‘ELISA-like’ assay [47]. Another found decreased global DNA methylation after a 10 uM BDE-47 in murine N2A cells (assessed by HPLC and arbitrary primed PCR). This decrease coincided with increased adipocyte differentiation (2.5–25 uM exposures) [48]. In a related effort to understand how endocrine-disrupting chemicals may be inducing adipocyte differentiation, investigators report that BDE-47 induces demethylation of several sites in the PPARγ promoter (a key adipongenic transcription factor) in COS7 and 3T3-L1 cells using Methylation-Sensitive High-Resolution Melting (MS-HRM) [49]. Complementing these in vitro findings, in vivo studies that perinatally exposed rodents to BDE-47 reported interesting findings from offspring of various ages. These include decreased expression of LINE1 RNA [50], decreased methylation of Mt-co2, L1Rn, Bdnf, and Nr3c1 [51], differentially methylated regions in sperm [52], and global DNA hypomethylation associated with behavioral deficits in both exposed wild-type and MeCP2-deficient female mice [53]. Another study in mice assessing liver carcinoma tissue after DE-71 (a commercial mixture of PBDEs) exposure found little effect on global DNA methylation but reports a gene body methylation decrease in Tbx3 and subsequent mRNA and protein upregulation [54]. While not directly assessing DNA methylation, a multigenerational study in zebrafish that exposed F0 animals to a PCB and PBDE mixture found disrupted behavior (hyper/hypoactivity) in F1–F4 larvae, as well as altered c-Fos expression (F1/2) and altered *Dmnt3ba* expression in all generations [55].

To our knowledge, only two reports exist that found no relationship between PBDE exposure and DNA methylation levels of any targets measured in those studies. One found no detectable decrease in methylation at the p53 promoter after 24 hours of exposure to low micromolar doses (1, 5, 10 umol/L) of BDE-47 in human neuroblastoma cells (SH-SY5Y), although activation of the p53 pathway in general was implicated in observed effects [56]. The other found no significant relationship between BDE-47 serum levels and global methylation as assessed by the luminometric methylation assay (LUMA) in samples from an elderly Swedish population. However, significant relationships were established for other persistent organic pollutants including PCBs and the dioxin OCDD [57]. Aside from these reports, the literature suggests a fairly consistent—but not necessarily linear—relationship between PBDE exposures and DNA methylation levels. It is possible that changes may vary from genomic region to region and may not always manifest an altered phenotype. Also, there is little evidence concerning direct cause–effect relationships between methylation changes and behavioral phenotypes. It will be both interesting and necessary to further refine understanding of the route by which PBDEs affect DNA methylation states—be it primarily by dysregulation of DNA methyltransferase expression, cellular metabolism, intracellular signaling pathways, etc. 

### 3.4. Impact on Chromatin—Histone Modifications to Chromatin Remodeling

Other reversible chemical modifications of chromatin include modifications to histone proteins that regulate chromatin structure and instruct remodeling processes, ultimately controlling gene expression [58,59]. Studies starting as early as 2003 reported mixed results on PBDEs, inducing altered chromatin by several measures (chromosomal integrity, chromatin density and localization). Exposure of multiple bacterial strains to BDE-99 did not induce mutagenicity or a detectable increase in the number of structural chromosomal aberrations, while exposure to the PCB mixture Aroclor® 1254 did [60]. This early study explicitly stated that the possibility of PBDEs acting through epigenetic mechanisms could not be ruled out, which, in retrospect, was prudent foresight. Two subsequent studies have also reported no increase in degraded chromatin, both in sperm—the first in the sperm of mice orally exposed to BDE-209 [61], the other in human samples of 153 men from the greater Montreal area, despite establishing a correlation between BDE-47 levels and decreased sperm concentration [62]. 

However, there have also been a few studies that do report chromatin disruption following PBDE exposure. One study found that 24 h nanomolar range exposures to several PBDEs (BDE-47/99/153/183/209) induced micronuclei formation during cytokinesis in MCF-7 cells, an indicator of chromosomal damage occurrence preceding cell division [63]. It has also been found that rat pups exposed to a single injection of BDE-153 at post-natal day 10 (PND10) exhibited behavioral dysfunction in a dose- and age-dependent manner one or two months later. Neurons in the CA3 region of the hippocampus of these rats were also found to be undergoing significantly increased rates of apoptosis, with chromatin condensed and localized to the nuclear membrane [64]. Most recently, it was reported that BDE-209 exposures reduced hESC differentiation (although total induction was still greater than 90%) and also led to chromosomal copy number variants (CNVs), as well as decreased expression of DNMT1/3A [65].

There is also some evidence specifically for PBDE-induced dysregulation of histones and histone-regulating proteins. In an effort to understand the carcinogenic potential of BDE-209, the first such study found that HEK293T cells exposed to micromolar range levels of the toxin exhibited altered expression of chromatin-regulating genes, specifically a histone gene cluster that the authors hypothesize could affect nucleosome properties [66]. In the same year, another group reported that exposure of the marine madaka (*Oryzias melastigma*) to BDE-47 led to sex-specific differential protein expression in male and female gonads, with several histone variants (H2b, H3.3, H3a, H2a) being down-regulated in male gonads [67]. Another study found that exposing maize (*Zea mays* L.) to BDE-47, and its metabolites 6-OH-BDE-47 and 6-MeOH-BDE-47, led to elevated levels of ROS and phospho-H2AX, likely in response to DNA damage. Interestingly, the hydroxylated metabolite produced the most severe effects [68]. Another study, primarily concerned with the relationship of PBDE exposure to reproductive health, exposed pregnant rats to BDE-47 from E8 to PND21. Male offspring were then assessed at PND120 for alterations in testes. It was found that exposed rats had smaller testes, decreased sperm production, and interestingly, an altered testes transcriptome and 4-fold decrease in protamine and transition gene expression (proteins responsible for histone-protamine exchange) [69]. Aside from these data that indicate potential disruptions of histone expression and nucleosome alteration, there are two studies that report PBDE-induced dysregulation of chromatin-regulating proteins. The first found that BDE-47 treatment downregulated SirT1 expression (a histone deacetylase) in the livers of mice due to NAD(+)-depletion [70]. Recently, we reported that chronic nanomolar range doses of a hydroxylated metabolite of BDE-47, 6-OH-BDE-47, altered NDD candidate gene expression, including several epigenetic regulators, particularly multiple components of the Brg1-associated factors (BAF) chromatin remodeling complex [71].

The potential importance of understanding the effects of PBDEs on chromatin dynamics cannot be understated given the fundamental importance of chromatin properties for regulating gene expression and thus cellular states. Going forward, it will be important for investigators to focus on identifying additional effects on chromatin while distinguishing those that are direct from indirect, hopefully allowing for elucidation of the underlying mechanism.

### 3.5. Other Affected Epigenetic Mechanisms (Non-Coding RNAs)

Non-coding RNAs—such as long non-coding RNA (lncRNA) and microRNA (miRNA)—can also act as epigenetic regulators [72,73]. Various PBDE exposures have been reported to alter expression of miRNAs, and one study described effects on expression of liver lncRNAs. The earliest study assessing miRNA expression as an endpoint following PBDE exposure utilized placental samples collected from the National Children’s Study. Among other associations established for PCB and heavy metal exposures, the study reported a positive association between BDE-209 and miR-188-5p expression and an inverse association for BDE-99 and let-7c [74]. Another group exposed hESCs in vitro to low doses of BDE-209 (1, 10, 100 nM) for 4 days, inducing apoptosis and downregulating pluripotency genes, particularly OCT4, in part by hypermethylation of the promoter and induction of miR-145/335 which repress OCT4. There was also generation of ROS and decreased superoxide dismutase (SOD2) expression. ROS and OCT4 effects were partially rescued by treatment with the antioxidant NAC [75]. An even more recent study employing human cells found that, after stimulating THP-1 macrophages with BDE-209 and LDL, there was dose-dependent repression of miRNA-21 which subsequently de-repressed toll-like receptor 4 expression (TLR4), enhancing TLR4-dependent lipid uptake [76,77,78]. 

In addition to these examples in humans, two rodent studies concerning non-coding RNAs in the liver have been published. The first found that BDE-47 exposure upregulates CYP3A1 in rat liver and that this upregulation is mediated by BDE-47-induced repression of miRNA-23b, which negatively regulates CYP3A1 mRNA via a 3’ UTR binding site [79]. The other study reported that conventional and gut-microbiome-depleted mice exhibit dysregulated lncRNA expression in liver tissue in response to both BDE-47 and BDE-99 exposure [80]. Interestingly, BDE-47 has also been found to induce dysregulation of novel miRNAs in exposed zebrafish larvae. Of particular interest is miR-735, which may play essential roles in larval sensory development, explaining previously observed BDE-47-induced disruption of zebrafish visual perception [81]. In the near future, a general model of PBDE-induced miRNA dysregulation may hopefully be established given the multiple intriguing examples already characterized.

## 4. Conclusions 

Considering this growing body of work documenting epigenetic dysregulation induced by PBDE exposure, there appear to be several central lines of evidence emerging from research done in various health contexts, including: adipocyte differentiation and obesity, reproductive health—of both sperm/testes and the placenta, carcinogenicity (especially thyroid related), and negative impacts on nervous system formation and function. It is becoming clear that many, if not all, of these various aspects of human health are impacted by PBDE-induced disruption of normal epigenetic states and mechanisms.

There are fairly consistent findings of a negative relationship between PBDE levels and DNA methylation from in vitro and non-human animal studies across varied cell/tissue types and methylation detection methods. However, the data from human samples is more difficult to interpret. Studies reporting effects on global DNA methylation levels inferred from representative regions have incongruent results, and evidence of alterations to methylation in the placenta are, likewise, not in direct agreement. However, this confoundment and the fact that human studies have so far been conducted across very different populations and models should only encourage further work on the topic, especially given indications from non-human animal and in vitro studies. It will be of great value if these types of studies can build on the tentatively established negative impact of PBDEs on methylation and begin to focus on understanding the mechanisms underlying the alterations, while continuing to clarify effects in human studies.

Compared to DNA methylation, the literature is poorer regarding the effects of PBDEs on other epigenetic mechanisms such as chromatin dynamics and expression of non-coding RNAs. However, some interesting ideas are beginning to emerge. While not yet well understood, PBDE-induced dysregulation of histones and chromatin regulators is an intriguing intersection for PBDEs and neurodevelopmental disorders, bolstered by the recent emergence of chromatin regulation as a major node of NDD risk [31,82]. Further, it is tempting to speculate that epigenetic effects of PBDE exposure may, generally, turn out to be a point of convergence for environmental and genetic factors that contribute to NDDs. If the effects of these compounds on targets such as DNA methylation, chromatin components and regulators, and non-coding RNA expression (all of which are mechanisms known to have roles in neurodevelopment and perhaps NDD etiology) can be further explored and resolved, one or more could very well turn out to be that link. This is of pressing importance, especially for neurodevelopmental disorders considering their explosive increase in prevalence, growing evidence for the involvement of PBDEs in their etiology, and the long elusive role of environmental factors in these devastating conditions.

Going forward, a major challenge for epigenetic PBDE research will be to assimilate new findings into the existing framework of PBDE toxicity that has been established from insights into other major impacted biological mechanisms. It will also be important to carefully consider nuanced aspects of exposures including tissue and sub-cellular localization, conduct more research on environmentally relevant doses and mixtures of PBDEs, further explore the prevalence and effects of their metabolites, and, to the extent that it is possible, integrate evidence generated across human and non-human studies (both in vitro and in vivo). This will be necessary in order to construct a more wholistic understanding of how these compounds impact cellular states and, ultimately, phenotypic outcomes. Hopefully, with continued research, we may eventually be able to explain how and to what extent these pervasive environmental pollutants are related to the numerous human health conditions that they appear to be contributing to. 

## Figures and Tables

**Figure 1 ijerph-16-02703-f001:**
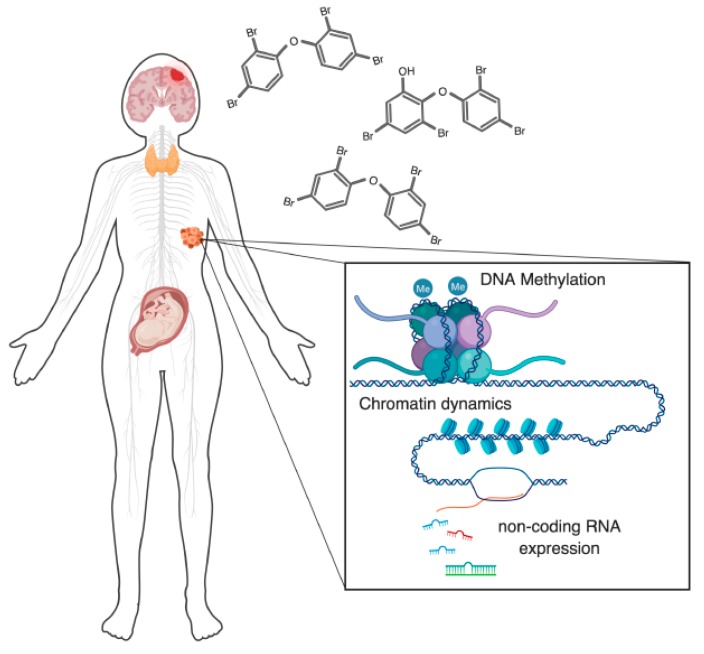
Polybrominated diphenyl ethers (PBDE) exposures affect epigenetic regulatory mechanisms at multiple levels, across multiple biological systems. There are several major aspects of human health that are of concern regarding PBDE toxicity that now have evidence for an involvement of dysregulated epigenetic regulation. These include: nervous system toxicity, disruption of thyroid hormone signaling, effects on the reproductive system (primarily on the placenta and testes), and oncogenic potential. One or more epigenetic components are known to be disrupted in each of these systems. In this review, we discuss these epigenetic regulators, their known modes of disruption by PBDEs, and the relationship of these disruptions to human health.

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
