# Peer review of "Epigenetic Effects of Polybrominated Diphenyl Ethers on Human Health"

_ijerph, 2019, doi:10.3390/ijerph16152703_

Round 1
Reviewer 1 Report
This interesting review summarized the whole body of in vitro, in vivo and human studies evaluating the epigenetic dysregulation induced by PBDEs, a widespread and persistent class of organic chemicals whose effects to human health need still to be elucidated. The review is well written, introduction provides sufficient elements to characterize the background, results of analyzed studies are properly and clearly reported, and references updated. In my opinion, the manuscript deserves to be published. I point out only minor revisions.
I recommend the authors to always write the full names of all acronyms used, sometimes they are missing (e.g., IGF, OCT4, NAC, LDL, NDD, DNMT1, E8 etc.). On the other hand, ROS as acronym was already written at line 124, thus its repetition at line 260 is redundant. When a name appears once in the text, you can avoid inserting the acronym. Otherwise, a list of abbreviations is recommended.
Lines 189-190: The authors mentioned in vivo studies, however in the following lines, they continued to refer to in vitro research. Could they explain this apparent inconsistency? Since from line 200 there is a paragraph on animal studies, hence the sentence at lines 189-190 could be eliminated.
Between lines 211 and 220, there is an excessive use of “though”.
Author Response
Response to Reviewer #1
This interesting review summarized the whole body of in vitro, in vivo and human studies evaluating the epigenetic dysregulation induced by PBDEs, a widespread and persistent class of organic chemicals whose effects to human health need still to be elucidated. The review is well written, introduction provides sufficient elements to characterize the background, results of analyzed studies are properly and clearly reported, and references updated. In my opinion, the manuscript deserves to be published. I point out only minor revisions.
We thank you for your positive comments about this review. Your minor concerns are well noted and responded to in the following lines.
I recommend the authors to always write the full names of all acronyms used, sometimes they are missing (e.g., IGF, OCT4, NAC, LDL, NDD, DNMT1, E8 etc.). On the other hand, ROS as acronym was already written at line 124, thus its repetition at line 260 is redundant. When a name appears once in the text, you can avoid inserting the acronym. Otherwise, a list of abbreviations is recommended.
A list of abbreviations has been added and the repetition at line 260 is removed. Thank you.
Lines 189-190: The authors mentioned in vivo studies, however in the following lines, they continued to refer to in vitro research. Could they explain this apparent inconsistency? Since from line 200 there is a paragraph on animal studies, hence the sentence at lines 189-190 could be eliminated.
Noted. The first line (189) was to switch over to animal research data from the human data in the previous paragraph. We apologize for not being clear. We have re-stated that first line to eliminate ambiguity.
Between lines 211 and 220, there is an excessive use of “though”.
We have revised those sentences and eliminated several ‘though’. Thank you.
Reviewer 2 Report
Overall comments: This manuscript addresses a very timely topic related to human health. The manuscript is well written and organized. However, I do have several comments for the authors’ consideration.
Specific comments:
· The manuscript needs a technical edit. I found numerous grammatical and editorial errors in all sections of the manuscript. Please consider correcting.
· Lines 29 and 40, Abbreviation for the United States Environmental Protection Agency, please either use “US EPA” or “EPA”, but not both in the same section of the manuscript.
· Line 36, delete the word “just”.
· Figure 1, it is not clear to me that this figure is needed to support the text. Both the text describing figure 1 and figure 1 are very generic. The text would probably be sufficient for the authors to make their point.
· Section 2.1, is extremely brief. As the authors noted, there has been much study of PBDEs for the last decade. In conducting a literature review in PubMed, I see many articles that would be relevant to this review paper that are not included. If this manuscript is truly going to be listed as a review, then it needs to be a review. This reviewer does not consider Section 2.1 complete.
· As a labeled review paper for IJERPH, it should have a methods section which details how the review was conducted and the papers were selected for inclusion in this manuscript under consideration. The authors need to document their process: did they use PRISMA or some other approach? Without this section, this reviewer deems the manuscript not worthy of publication.
· Lines 56-74, the early history of PBDEs is very interesting. However, it is unclear how the fact that PBDEs were found in marine sponges and red algae relates to current ubiquitousness. And, would it be more useful to describe how ubiquitous PBDEs really are?
· Line 171, the acronym for the CHAMACOS study needs to be corrected. There are many other human studies where PBDEs have been measured, so it is unclear to this reader why CHAMACOS is the only study highlighted.
· Line 345, please delete the “w” from the word holistic.
Author Response
Response to Reviewer #2
Overall comments: This manuscript addresses a very timely topic related to human health. The manuscript is well written and organized. However, I do have several comments for the authors’ consideration.
We thank you for considering this topic timely and finding this manuscript well written and organized. Your comments are constructive and we have done our best to considered them carefully and respond positively. Please see our point-by-point responses below.
Specific comments:
· The manuscript needs a technical edit. I found numerous grammatical and editorial errors in all sections of the manuscript. Please consider correcting.
· Lines 29 and 40, Abbreviation for the United States Environmental Protection Agency, please either use “US EPA” or “EPA”, but not both in the same section of the manuscript.
Corrected to EPA. Thank you.
· Line 36, delete the word “just”.
Deleted. Thank you.
· Figure 1, it is not clear to me that this figure is needed to support the text. Both the text describing figure 1 and figure 1 are very generic. The text would probably be sufficient for the authors to make their point.
Respectfully, we disagree with your comment here. We intend the figure to supplement the text in the format of an visual abstract. Visual abstracts are often appreciated by uninitiated readers new to the field. We also noted that the other reviewer had no objection to the figure. We are therefore requesting the retention of the figure as is.
· Section 2.1, is extremely brief. As the authors noted, there has been much study of PBDEs for the last decade. In conducting a literature review in PubMed, I see many articles that would be relevant to this review paper that are not included. If this manuscript is truly going to be listed as a review, then it needs to be a review. This reviewer does not consider Section 2.1 complete.
Our review is by no means is an exhaustive review of PBDE-related threats to overall human health. In the introductory section 2.1, our intention is to broadly introduce the reader to general PBDE-related threats to overall human health and then focus on a particularly important aspect of the same: neurodevelopment disorders (hence titled, ‘Human exposure to PBDEs and effects on human health, especially neurodevelopmental disorders’). In lines 82-84, we have also indicated that a curious reader will be directed to further reading on other topics of human health, which is then done by referencing reviews and meta-studies throughout the segment and the essay.
· As a labeled review paper for IJERPH, it should have a methods section which details how the review was conducted and the papers were selected for inclusion in this manuscript under consideration. The authors need to document their process: did they use PRISMA or some other approach? Without this section, this reviewer deems the manuscript not worthy of publication.
We have now included a methods section. Please see lines 362-389. Thank you.
· Lines 56-74, the early history of PBDEs is very interesting. However, it is unclear how the fact that PBDEs were found in marine sponges and red algae relates to current ubiquitousness. And, would it be more useful to describe how ubiquitous PBDEs really are?
It was never our intention to suggest that naturally occurring PBDEs relate to the latter’s current ubiquitousness. We wanted to point out that PBDEs have both natural and anthropogenic origins. We have re-worded this section to make it clearer for our readers (see lines 56-60).
· Line 171, the acronym for the CHAMACOS study needs to be corrected. There are many other human studies where PBDEs have been measured, so it is unclear to this reader why CHAMACOS is the only study highlighted.
Acronym is corrected. Thank you. Two additional studies (Korean and Boston) have also been highlighted in this section.
· Line 345, please delete the “w” from the word holistic.
Corrected. Thank you.
Round 2
Reviewer 2 Report
I have reviewed the revised manuscript and the response to comments document. I commend the authors on providing adequate responses to the comments and improving the manuscript. I think the revised manuscript is improved and should be considered for publication in IJERPH.